# *Helicobacter pylori* Strains Isolated from Raw Poultry Meat in the Shahrekord Region, Iran: Frequency and Molecular Characteristics

**DOI:** 10.3390/genes14051006

**Published:** 2023-04-28

**Authors:** Sepehr Asadi, Ebrahim Rahimi, Amir Shakerian

**Affiliations:** 1Department of Food Hygiene, Shahrekord Branch, Islamic Azad University, Shahrekord P.O. Box 166, Iran; 2Research Center of Nutrition and Organic Products, Shahrekord Branch, Islamic Azad University, Shahrekord P.O. Box 166, Iran

**Keywords:** *Helicobacter pylori*, raw poultry flesh, antibiotic resistance, genotyping

## Abstract

Even though *Helicobacter pylori* (*H. pylori*) is a serious pathogen, its origin is unknown. Poultry (chicken, turkey, quail, goose, and ostrich) is consumed as a regular protein source by many people across the world; therefore, sanitary ways of delivering poultry for food are important for global health. As a result, the distribution of the virulence genes *cagA*, *vacA*, *babA2*, *oipA*, and *iceA* in *H. pylori* isolates in poultry meat, as well as their antibacterial resistance, was investigated. A Wilkins Chalgren anaerobic bacterial medium was used to cultivate 320 samples of raw poultry meat. Disk diffusion and multiplex-PCR were used to investigate both antimicrobial resistance and genotyping patterns. *H. pylori* was found in 20 of 320 (6.25 %) raw chicken meat samples. The highest incidence of *H. pylori* was found in chicken raw meat (15%), whereas no isolate was recovered from goose or quail raw meat (0.00%). Resistance to ampicillin (85%), tetracycline (85%), and amoxicillin (75%) were the most commonly detected in the tested *H. pylori* isolates. The percentage of *H. pylori* isolates with a multiple antibiotic resistance (MAR) index value of more than 0.2 was 17/20 (85%). The most prevalent genotypes detected were *VacA* (75%), *m1a* (75%), *s2* (70%) and *m2* (65%), and *cagA* (60%). The most typically detected genotype patterns were *s1am1a* (45 %), *s2m1a* (45 %), and *s2 m2* (30%). *babA2*, *oipA*+, and *oipA*- genotypes were found in 40%, 30%, and 30% of the population, respectively. In summary, fresh poultry meat was polluted by *H. pylori*, with the *babA2*, *vacA*, and *cagA* genotypes being more prevalent. The simultaneous occurrence of *vacA*, *cagA*, *iceA*, *oipA*, and *babA2* genotypes in antibiotic-resistant *H. pylori* bacteria raises a serious public health concern regarding the consumption of raw poultry. Future research should evaluate antimicrobial resistance among *H. pylori* isolates in Iran.

## 1. Introduction

The genus Helicobacter is a group of Gram-negative bacteria known for their helical, curved, or sometimes coccoid (spherical) shape. Helicobacter bacteria are generally small, with a length ranging from 2 to 4 μm and a width ranging from 0.5 to 1.0 μm [1,2]. Helicobacter bacteria can be classified into two main categories based on their preferred location of colonization in the body. The first category is the stomach Helicobacter, which colonize the gastric mucosa (the lining of the stomach) and include species such as *Helicobacter pylori* (*H. pylori*). These bacteria are associated with a range of gastrointestinal diseases, including gastritis, peptic ulcers, and stomach cancer [3,4,5]. The second category is enterohepatic Helicobacter, which colonize the liver, biliary tract, and intestine. These bacteria include species such as Helicobacter hepaticus and Helicobacter bilis, and are associated with a range of conditions such as hepatitis, cholecystitis, and inflammatory bowel disease [6]. Recently, Helicobacter have been referred to as zoonotic bacteria [7].

Poultry (chicken, turkey, quail, goose, and ostrich) is a popular protein source in countries across the globe [8,9]. The main reasons for the interest in poultry meat are its relatively low production costs and the absence of cultural and religious restrictions on its consumption [10]. Contamination of poultry meat can occur at various stages of its slaughter and processing [11,12]. Studies have shown that *H. pylori* can survive in the gastrointestinal tract of chickens, and these bacteria can be present in their feces [13]. If proper hygiene and sanitation practices are not followed during slaughter and processing, these bacteria can contaminate the poultry meat. Consumption of poultry meat contaminated with *H. pylori* may increase the risk of gastrointestinal infections in humans [5]. Sanitary delivery techniques for food-producing chickens are therefore essential for public health [14].

The emergence of antibiotic-resistant *H. pylori* strains has become a global issue, and many published studies reveal that *H. pylori* strains isolated from food sources, as well as clinical specimens, show a high rate of resistance against various types of antimicrobial medications, including macrolide, fluoroquinolones, metronidazole, tetracyclines, penicillin, aminoglycosides, and sulfonamides [15,16,17]. *H. pylori* resistance to antibacterial treatments has become a global issue, and many published studies have shown that both clinical and foodborne strains of *H. pylori* exhibited a high rate of resistance to various antibiotics [18,19,20,21]. While *H. pylori* is a frequently reported pathogen in humans and poultry, its source is still unclear [22]. *H. pylori* pathogenesis is linked to virulence genes only found in certain groups. Several virulence genes in *H. pylori* isolates have been reported, including vacuolating cytotoxin A (*VacA*) [23], cytotoxin-associated A (*CagA*) [23], restriction endonuclease A (*IceA*) [24], outer inflammatory protein A (*OipA*) [25], and blood-group antigen-binding adhesin (*BabA2*) [25]. Among the most efficient ways for studying relationships between *H. pylori* strains from diverse samples is genotyping using the virulence genes (*cagA*, *vacA*, *babA2*, *oipA*, and *iceA*) [15].

*H. pylori* is known to be prevalent in Iran. According to a systematic review and meta-analysis published in 2020, the overall prevalence of *H. pylori* infection in Iran was found to be 54% [26]. The prevalence varied across different regions of Iran, with higher rates reported in the northern and western regions. Several risk factors have been associated with *H. pylori* infection in Iran, including poor sanitation, low socioeconomic status, and low education level. Contaminated water and food, including poultry meat, have also been identified as potential sources of infection [27]. However, there have been insufficient published studies on antibiotic resistance of *H. pylori* obtained from edible and inedible raw poultry (chicken, turkey, quail, goose and ostrich) in Iran. Thus, the present study aims to characterize *H. pylori* strains by using virulence genes as markers and to assess the resistance profiles of the isolates.

## 2. Materials and Methods

### 2.1. Sampling

From April to July 2020, 320 raw poultry samples, including chicken (*n* = 60), turkey (*n* = 55), quail (*n* = 65), goose (*n* = 65), and ostrich (*n* = 75), were randomly collected from a slaughterhouse in the Shahrekord region, Iran. The sample size was determined using the Schwartz [18] formula $n=\frac{z^{2}\;\times \;p\;\times \;q}{d^{2}}$, where *n* = required sample size; *p* = prevalence; *p* = 0.54; q = 1 − *p*; z = confidence level according to the centered reduced normal distribution (for a confidence level of 95%, z = 1.96); and d = margin of error tolerated for this study is 0.05. Therefore, *n* = 382. Samples (40 g) were collected from breast, livers, and gizzards, including the jejunum, cecum, and colon. All samples were stored in a specific sterile Ziploc bag (Sc Johnson) that was water-resistant and transported in a cool chain (at 4 °C) to the laboratory within 2 h after collection.

### 2.2. Colony Morphology, Gram-Staining and Biochemical Analysis of H. pylori

For the isolation of *H. pylori*, 25 g of each sample was suspended in 225 mL of Brucella broth supplemented with 5% sheep defibrinated blood and DENT selective supplement (Oxoid, Hampshire, UK). After incubation at 37 °C for 48 h under a microaerophilic condition using BBL GasPak™ jars (Becton Dickinson, Franklin Lakes, NJ, USA), supplemented with CampyGen bags (Oxoid, Hampshire, UK) 100 µL of the mixture was inoculated onto Columbia blood agar base (Oxoid, Hampshire, UK), supplemented with 5% sheep blood and DENT selective supplements, and then incubated for 5–7 days under a microaerophilic condition. Suspected colonies were further identified using Gram staining (Gram-negative for *Helicobacter* spp.), oxidase (positive for *Helicobacter* spp.), urease (positive for *H. pylori*), and nitrate reduction (positive for *H. pullorum*) tests.

### 2.3. Genotypical Identification of H. pylori by 16S rRNA-Based PCR Confirmation

Subculturing was carried out using Wilkins Chalgren anaerobe medium. Genomic DNA was extracted from bacteria using a DNA extraction kit following the manufacturer’s instructions (Cinna-colon, Iran). The extracted DNA’s quality (A260/A280) and quantity (ng/mL) were then assessed using NanoDrop spectrophotometer (Thermo Scientific, Waltham, MA, USA). The *Helicobacter* genus was identified by targeting 16S rRNA (Table 1). The oligonucleoide sequence was approved by the Lactofeed Biotech Group (Iran). A PCR thermal cycler (Eppendorf Co., Hamburg, Germany) was used to execute the polymerase chain reaction (PCR) according to the conditions described by the author for the used gene sequences [28]. Segments of the 16S rRNA gene were analyzed by PCR amplification. PCR conditions were as follows: an initial denaturation (94 °C, 5 min), followed by 40 cycles of denaturation (94 °C, 30 s), annealing (50 °C, 1 min), and extension (72 °C, 2 min), with a final extension (72 °C, 8 min). *H. pylori* ATCC 700392 strain and distilled water were used as positive and negative controls, respectively.

### 2.4. H. pylori Antibiotic Susceptibility Pattern

There seem to be no generally recognized standardized methods for checking *H. pylori* antibiotic susceptibilities; therefore, the procedures shown in this research were based on Ranjbar et al. [2] and Performance Standards for Antibiotic Susceptibility Testing, Clinical and Laboratory Standards Institute, 30th ed CLSI supplement M100. To inoculate Muller–Hinton agar plates, bacterial solutions were diluted to the 0.5 Mcfarland (equal to 1–2 × 10^8^ CFU/mL). The current study employed antibiotic discs with varied doses to investigate the in vitro susceptibility of *H. pylori* isolates to antibiotics routinely used for treating *H. pylori* [29,30]. Antimicrobial discs (amoxicillin (10 μg), ampicillin (10 μg), metronidazole (5 μg), streptomycin (10 μg), cefsulodin (30 μg), erythromycin (5 μg), levofloxacin (5 μg), trimethoprim (25 μg), furazolidone (1 μg), clarithromycin (2 μg), rifampin (30 μg), tetracycline (30 μg), and spiramycin (100 μg) (Mast, UK) were used, and the plates were incubated at 35 °C for 16–18 h under anaerobic conditions. The standard technique was used to assess and analyze the inhibition zone induced by each antibiotic following European Committee on Antimicrobial Susceptibility Testing instructions (2020). *H. pylori* ATCC 26695 and ATCC 43504 were used as quality management isolates. Ayandele et al. (2020) [31] proposed the following calculation to calculate the Multiple Antibiotic Resistance score (MAR) index of each strain:

**Table 1 genes-14-01006-t001:** Oligonucleotide sequences, product length and cycling conditions of *H. pylori* virulence genotypes.

Target Gene		Oligonucleotide Sequence (5′–3′)	Size (bp)	Annealing Temperature (°C)	Authors
**16S rRNA**		F: CTATGACGGGTATCCGGC R: ATTCCACCTACCTCTCCCA	375	53	[28]
** *VacA* **	** *s1a* **	F: CTCTCGCTTTAGTAGGAGC R: CTGCTTGAATGCGCCAAAC	213	64	[32]
	** *s1b* **	F: AGCGCCATACCGCAAGAG R: CTGCTTGAATGCGCCAAAC	187	64	[32]
	** *s1c* **	F: CTCTCGCTTTAGTGGGGYT R: CTGCTTGAATGCGCCAAAC	213	64	[32]
	** *s2* **	F: GCTAACACGCCAAATGATCC R: CTGCTTGAATGCGCCAAAC	199	64	[32]
	** *m1a* **	F: GGTCAAAATGCGGTCATGG R: CCATTGGTACCTGTAGAAAC	290	64	[32]
	** *m1b* **	F: GGCCCCAATGCAGTCATGGA R: GCTGTTAGTGCCTAAAGAAGCAT	291	64	[32]
	** *m2* **	F: GGAGCCCCAGGAAACATTG R: CATAACTAGCGCCTTGCA	352	64	[32]
** *CagA* **	** *CagA* **	F: GATAACAGCCAAGCTTTTGAGG R: CTGCAAAAGATTGTTTGGCAGA	300	56	[32]
** *IceA* **	** *IceA1* **	F: GTGTTTTTAACCAAAGTATC R: CTATAGCCATYTCTTTGCA	247	56	[33]
	** *IceA2* **	F: GTTGGGTATATCACAATTTAT R: TTCCCTATTTTCTAGTAGGT	229	56	[33]
** *OipA* **		F: GTTTTTGATGCATGGGATTT R: GTGCATCTCTTATGGCTTT	401	56	[33]
** *BabA2* **		F: CCAAACGAAACAAAAAGCGT R: GCTTGTGTAAAAGCCGTCGT	105	57	[34]

MAR index = number of antimicrobial drugs to which the bacterium is resistant/total number of antimicrobial drugs.

### 2.5. Genotyping Analysis

Multiplex-PCR was used to determine the prevalence of *cagA*, *vacA*, *babA2*, *oipA*, and *iceA* alleles [25,35,36,37]. The primers and PCR conditions used to genotype *cagA*, *vacA*, *babA2*, *oipA*, and *iceA* alleles are listed in Table 1. Initially, all samples were subjected to pre-tests to determine the best reaction time, temperature, and volume. In all PCR operations, a programmed DNA thermo-cycler was employed. Positive and negative controls were *H. pylori* standard strains (ATCC 43504) and PCR-grade water, respectively. A total volume of 25 µL was used, consisting of 5 µL of deoxy–nucleoside triphosphate mix, 2.5 µL of 10× PCR buffer, 0.25 µL of the primer, and 1 µL of the DNA template. Ethidium bromide was used to dye ten microliters of PCR product electrophoresed in a 2 percent agarose gel in 1× TBE buffer at 80 V for 30 min. For image processing, UVI doc gel documentation devices (Grade GB004, Jencons PLC, London, UK) were used.

### 2.6. Analytical Statistics

The International Business Machines Corporation (IBM) Statistical Package for Social Sciences (SPSS) software (version 20.0 for Windows) was used to conduct the statistical study. The information was given in the form of a mean, standard deviation, or percentages.

## 3. Results

### 3.1. Helicobacter spp. Prevalence in Poultry Based on Morphological and Biochemical Analysis

The presence of *H. pylori* was evaluated in 320 samples of poultry meat (Table 2). *Helicobacter spp*. were found in 20 of 320 (6.25%) poultry meat specimens. According to our findings, nine (15.00 %) chicken samples, seven (12.72 %) turkey samples, zero (0 %) quail samples, zero (0 %) goose samples, and four (5.33 %) ostrich samples were all contaminated with *Helicobacter* spp.

### 3.2. Identification of H. pylori with PCR Targeting the 16SrRNA Gene

The 16SrRNA gene PCR amplification was used to confirm all of the strains. The electrophoretically displayed PCR results from 20 *H. pylori* identified from 320 poultry samples and *H. pylori* are presented for all cases (100%). According to PCR results, all 20 (100%) isolates belonged to *H. pylori*. The largest prevalence of *H. pylori* bacteria was found in chicken (15.00%) and turkey (12.72%) meat samples, while no prevalence was found in quail and goose (0.00%).

### 3.3. H. pylori Sensitivity to Antibiotics and the MAR Index

Antimicrobial resistance profiles of *H. pylori* isolates recovered from various origins collected are depicted in Table 3. Antimicrobial resistance was found to be most common against ampicillin (85%), tetracycline (85%), and amoxicillin (75%). *H. pylori* isolates also had the lowest rate of resistance to furazolidone (5%), spiramycin (30%), cefsulodin (30%), and levofloxacin (30%). Furthermore, resistance to metronidazole (50%) and streptomycin (50%) was common, as it was resistant to erythromycin and rifampicin (40%), as well as trimethoprim and clarithromycin (35%). The results show that 17/20 (85%) of the *H. pylori* isolates obtained from poultry samples were resistant to at least three antibiotics from different classes (multi-drug resistant or MDR isolates). The average MAR index of the 20 H. pylori isolates recovered from poultry meat was 0.622 (Table 4). Seventeen of the twenty *H. pylori* isolates tested positive for antibiotic resistance (MDR phenotype), with MAR indexes varying from 0.230 to 1. Isolates 1 and 2 were highly resistant to all antibacterial agents (MAR index of 1.0), whereas isolates 3–5 were resistant to 12 of the 13 tested antibiotics (MAR index of 0.923). The MAR index for strains 6–7 was 0.846. The MAR scores for isolates 8–17 ranged from 0.23 to 0.769, and isolates 19 and 20 had the lowest MAR score (0.076). The number of *H. pylori* isolates with a MAR value more than 0.2 was 17/20 (85%); the frequency with a MAR value less than 0.2 was 3/20 (15%). We therefore conclude that *H. pylori* is extremely resistant to numerous evaluated antibacterial drugs and has large MAR index values.

### 3.4. Genotype Distribution among H. pylori Isolates Obtained from Various Origins or Poultry Samples

The genotype distribution of *H. pylori* isolates recovered from various samples is shown in Table 5. The most commonly found genotypes among the *H. pylori* bacteria isolates recovered from poultry raw meat were *vacA s1a* (75 %), *m1a* (75 %), s2 (70 %) and m^2^ (65 %), and *cagA* (60 %). The *H. pylori* isolates identified from several sorts of poultry samples with the lowest frequency were *vacA s1c* (5%) and *iceA2* (15%). *vacAs1b*, *vacAm1b*, and *oipA* genes were also found in 25% of *H. pylori* strains from various poultry specimens. *iceA1* and *babA2* genes were distributed in 40% of the population.

### 3.5. Genotyping Patterns of H. pylori Strains

The genotyping frequency of *H. pylori* isolates recovered from the various poultry samples is shown in Table 6. The most commonly found *vacA* genotypes of *H. pylori* bacteria originating from the different poultry samples were *s1am1a* (45%), *s2m1a* (45%), and *s2 m2* (30%). *babA2*, *oipA*+, and *oipA*- genotypes were seen in 40%, 30%, and 30%, respectively. We detected that *iceA1/iceA2* genotyping was present in 10% of *H. pylori* isolates. Among the diverse genotyping profiles of *H. pylori* isolates, *s1cm1b* (0%), *s1 cm2* (5%), *s1cm1a* (5%), and *cagA*+ (5%) exhibited the lowest frequency. The distribution of other genotypes, including *s1am1b* (15%), *s1 am2* (25%), *s1bm1a* (15%), *s1bm1b* (10%), s1bm2 (10%), *s2m1b* (15%), *cagA*- (25%), *babA2*+ (25%) and *iceA1/iceA2* (10%), was moderate.

## 4. Discussion

Data on the epidemiological role of poultry as a vehicle of *H. pylori* are very scarce. The present study aimed to investigate the frequency and molecular characteristics of *H. pylori* strains isolated from raw poultry meat. In the present study, 20 (6.25 %) *H. pylori* strains were recovered from 320 poultry samples, indicating that this bacterium poses a risk to humans. Although the root cause of such a finding is unknown, the cross-contamination of poultry meat is considered a major source of *H. pylori* infection in the poultry meat industry. The three primary operations that may increase the incidence of *H. pylori* infection include cutting, keeping, and shipping poultry meat. Furthermore, in a separate study, Ranjbar et al. [2] found that *H. pylori* can survive in water [2]. Therefore, an additional cause of *H. pylori* occurrence in the obtained poultry sample is the use of contaminated water in the meat industry. Furthermore, contaminated slaughtering personnel and equipment, such as blades, may contribute to a higher prevalence of this pathogen [38]. Generally, our findings are similar to those of Meng et al. (2008), who used multiplex-PCR to analyze 11 fresh chicken specimens (total chicken including skin), identifying that 4 (36%) were *H. pylori*-positive, although the prevalence detected in our study (6.25%) was significantly smaller. *H. pylori* is also a foodborne organism that may be transferred to humans, according to these investigators [39]. *H. pylori* is one of the main organisms that causes food poisoning, especially in raw meat products.

El Dairouty et al. (2016) reported that 5% of ground beef, raw chicken, and sandwich meat specimens tested positive for *H. pylori* [40].

Genomic approaches have subsequently been employed by several studies to discover the various genotypes of *H. pylori*, which also are strongly connected to its distribution. Multiplex-PCR is a commonly used test for genotyping and identifying homologous genes in *H. pylori* isolates obtained from clinical specimens [40,41]. The 16S rRNA gene was employed as a reference gene in this investigation. This gene was detected in all isolates, indicating that the 16S rRNA gene is a good candidate for identifying different *H. pylori* isolates. Similar findings were achieved by Piri-Gharaghie (2018) [7] and El Dairouty et al. (2016) [29,40]. According to these scientists, the 16S rRNA genetic code is a unique gene for recognizing bacterial species recovered from specimens when contrasted to other reference genes. Proliferation and cell wall formation in *H. pylori* are is dependent on the 16S rRNA gene. As a result, this gene has been widely used to diagnose *H. pylori* infections [41]. Our study looks at the incidence of the virulence genes, *iceA*, *babA2*, *oipA*, *vacA*, and *cagA*. In *H. pylori* isolates collected from edible and non-edible tissues from the poultry meat industry, the *IceA* (27.5 %), *babA2* (40 %), *OipA* (25 %), *vacA* (48.57 %), and *cagA* (60 %) genes were all found. As a result, certain virulence genes, notably *cagA*, were found in larger numbers in commercial poultry meat. The major impediments of *H. pylori* in the human digestive system are thought to be increased by these genotypes. Bibi et al. (2017) earlier hypothesized a relationship between the existence of the *H. pylori babA2*/*cagA*+/*vacAs1* genotype and the prevalence of gastroenteritis, stomach carcinoma, and ulcerative colitis [15]. In *H. pylori* isolates recovered from clinical specimens of human and animal populations, a high incidence of *vacA*, *cagA*, *iceA1*, *oipA*, and *babA2* genotypes was also found [15,42,43]. Moreover, *H. pylori* isolates recovered from different sorts of dietary specimens have shown a significant frequency of these genes [44,45]. Previous studies have linked the *H. pylori* genotypes *vacA*, *cagA*, *iceA*, *oipA*, and *babA2* to interleukin−8 and cytotoxin exudation, attachment to gastric epithelial cells, increase in the frequency of inflammatory impact, vacuolization, apoptosis process in gastric epithelial cells, stomach ulcers ulceration, increased intense neutrophilic incursion [43,44,45,46,47]. The consumption of fresh poultry meat infected with virulent isolates of *H. pylori* could increase duodenum ulcers, gastric epithelium shrinkage, and stomach carcinoma because *the H. pylori* strain in this experiment carried the *vacA*, *cagA*, *iceA*, *oipA*, and *babA2* genes. Furthermore, certain *H. pylori* strains tested positive for multiple genotypes at the same time, indicating that they are potentially more harmful [21].

Another noteworthy result in the ongoing investigation is the high prevalence of antibiotic resistance among *H. pylori* isolates, *which have frequent* resistance to antimicrobials ampicillin (85%), tetracycline (85%), and amoxicillin (75%). Our results are consistent with those of Hamada et al. (2018), who found high levels of antimicrobial resistance to amoxicillin, penicillin, oxytetracycline, nalidixic acid, ampicillin and norfloxacin in *H. pylori* isolates recovered from chicken meat, liver, and gizzards [13]. Similarly, Mashak et al. (2020) found *H. pylori* strains resistant to tetracycline, erythromycin, levofloxacin and amoxicillin in raw meat [48]. In addition, *H. pylori* bacteria highly resistant to ampicillin, erythromycin, amoxicillin, tetracycline and clarithromycin were found in samples of meat products by Gilani et al. (2017) [38]. Recently, Elrais et al. (2022) reported the isolation of multidrug resistant *H. pylori* strains from poultry chicken meat samples [49]. Furthermore, epidemiologic studies in different countries found that *H. pylori* isolates in healthcare specimens had a high level of resistance to antibiotics such as metronidazole, ampicillin, tetracycline, and amoxicillin, which is consistent with our results [44,45,46,47,50,51]. With a MAR index, 85 % of the *H. pylori* isolates were resistant for three or more antibiotic medicines employed in the study, indicating a large chance of infection in poultry. Antibiotic resistance may have become more common as a result of the nonselective use of such antibacterial medicines, according to our findings. Many researchers have looked at the incidence of *H. pylori* resistance to multiple antibiotics, but some researchers ran into problems, notably with the number of isolates studied [52,53]. A low resistance rate of *H. pylori* isolates to metronidazole (50%), streptomycin (50%), erythromycin (40%), rifampin (40%), trimethoprim (35%), and clarithromycin (35%) was also found in our study. These findings might be attributed to the antibiotic medications being prescribed less often. There has been some speculation about a link between virulence genes and antibiotic resistance. According to research conducted in Ireland in 2009, the lack of *cagA* could be a potential risk for acquiring metronidazole sensitivity [54]. Other research has linked clarithromycin susceptibility changes to the less pathogenic *vacA* genotypes [55]. Some other studies identified a link between *cagE* and *vacA S1,* as well as clarithromycin and metronidazole susceptibility [56], whereas others reported no link between *cagA* or *vacA* and susceptibility [33]. As a result, it is crucial to determine whether there is a link between the existence of pathogenic indicators and antimicrobial resistance within *H. pylori* isolates.

## 5. Conclusions

In conclusion, the present study shows cases of H. pylori contamination of raw poultry meat collected from slaughterhouses. Many isolates were also found resistant to antibiotics, including metronidazole and streptomycin, erythromycin, rifampin, trimethoprim, and clarithromycin, often used as antibacterial treatments in a clinical setting. Various virulence profiles were found, with some isolates carrying several virulence genes. Therefore, sanitary measures in slaughterhouses and butchers are critical for reducing the risk of H. pylori infection from poultry meat spreading to humans. Future genome sequencing of isolates will help explain the mechanism of resistance observed and establish the genetic relatedness of the isolates for a better understanding of their epidemiology.

## Figures and Tables

**Table 2 genes-14-01006-t002:** Prevalence of *H. pylori* in different types of raw poultry meat samples.

Raw Meat Samples	No Samples Collected	*n* (%) of *H. pylori* Positive Samples	*H. pylori 16SrRNA* PCR Confirmation (%)
Chicken	60	9 (15.00)	9 (15.00)
Turkey	55	7 (12.72)	7 (12.72)
Quebec	65	0	0
Goose	65	0	0
Ostrich	75	4 (5.33)	4 (5.33)
Total	320	20 (6.25) (95%)	20 (6.25) (95%)

**Table 3 genes-14-01006-t003:** Antibiotic resistance pattern of *H. pylori* strains isolated from different types of raw poultry meat samples.

Type of Raw Milk Samples (Number of *H. pylori* Strains)	Number (%) Isolates Resistant to Each Antibiotic												
	AM10^a^	Met5	ER5	CLR2	AMX 10	Tet30	Lev5	S10	RIF30	Cef30	TRP25	FZL1	Spi100
Chicken (9)	8 (88.88)	6 (66.66)	4 (44.44)	4 (44.44)	7 (77.77)	7 (77.77)	3 (33.33)	2 (22.22)	3 (33.33)	2 (22.22)	3 (33.33)	2 (22.22)	3 (33.33)
Turkey (7)	5 (71.42)	2 (28.57)	3 (42.85)	2 (28.57)	5 (71.42)	7 (100)	2 (28.57)	7 (100)	4 (57.14)	4 (57.14)	3 (42.85)	3 (42.85)	3 (42.85)
Ostrich (4)	4 (100)	2 (50)	1 (25)	1 (25)	3 (75)	3 (75)	1 (25)	1 (25)	1 (25)	–	1 (25)	–	–
Total (20)	17 (85)	10 (50)	8 (40)	7 (35)	15 (75)	17 (85)	6 (30)	10 (50)	8 (40)	6 (30)	7 (35)	5 (25)	6 (30)

**Table 4 genes-14-01006-t004:** Antimicrobial resistance profile of *H. pylori* strains (*n* = 20).

No.	Antimicrobial Resistance Profile	MAR Index
1	AM10, Met5, ER5, CLR2, AMX10, Tet30, Lev5, S10, RIF30, Cef30, TRP25, FZL1, Spi100	1
2	AM10, Met5, ER5, CLR2, AMX10, Tet30, Lev5, S10, RIF30, Cef30, TRP25, FZL1, Spi100	1
3	AM10, Met5, ER5, CLR2, AMX10, Tet30, Lev5, S10, RIF30, Cef30, TRP25, FZL1	0.923
4	AM10, Met5, ER5, CLR2, AMX10, Tet30, Lev5, S10, RIF30, Cef30, TRP25, FZL1	0.923
5	AM10, Met5, ER5, CLR2, AMX10, Tet30, Lev5, S10, RIF30, Cef30, TRP25, FZL1	0.923
6	AM10, Met5, ER5, CLR2, AMX10, Tet30, Lev5, S10, RIF30, Cef30, TRP25	0.846
7	AM10, Met5, ER5, CLR2, AMX10, Tet30, Lev5, S10, RIF30, Cef30, TRP25	0.846
8	AM10, Met5, ER5, CLR2, AMX10, Tet30, Lev5, S10, RIF30, Cef30	0.769
9	AM10, Met5, ER5, CLR2, AMX10, Tet30, Lev5, S10, RIF30, Cef30	0.769
10	AM10, Met5, ER5, CLR2, AMX10, Tet30, Lev5, S10, RIF30, Cef30	0.769
11	AM10, Met5, ER5, CLR2, AMX10, Tet30, Lev5, S10, RIF30	0.692
12	AM10, Met5, ER5, CLR2, AMX10, Tet30, Lev5, S10	0.615
13	AM10, Met5, ER5, CLR2, AMX10, Tet30, Lev5, S10	0.615
14	AM10, Met5, ER5, CLR2, AMX10, Tet30, Lev5	0.538
15	AM10, Met5, ER5, CLR2, AMX10	0.384
16	AM10, Met5, ER5, CLR2	0.307
17	AM10, Met5, ER5	0.230
18	AM10, Met5	0.153
19	AM10	0.076
20	AM10	0.076
**Average**	**0.622**	

**Table 5 genes-14-01006-t005:** Distribution of genotypes amongst the *H. pylori* strains isolated from different types of raw poultry meat samples.

Type of Raw Milk Samples (Number of *H. pylori* Strains)	Number (%) Isolates Harbor Each Genotype											
	*VacA*							*CagA*	*IceA*		*OipA*	*BabA2*
	*s1a*	*s1b*	*s1c*	*s2*	*m1a*	*m1b*	*m2*		*IceA1*	*IceA2*		
Chicken (9)	7 (77.77)	3 (33.33)	1 (11.11)	6 (66.66)	7 (77.77)	3 (33.33)	6 (66.66)	6 (66.66)	4 (44.44)	2 (22.22)	3 (33.33)	4 (44.44)
Turkey (7)	6 (85.71)	2 (28.57)	-	6 (85.71)	6 (85.71)	1 (14.28)	5 (71.42)	5 (71.42)	3 (42.85)	1 (14.28)	2 (28.57)	3 (42.85)
Ostrich (4)	2 (50)	-	-	2 (50)	2 (50)	1 (25)	2 (50)	1 (25)	1 (25)	-	-	1 (25)
Total (20)	15 (75)	5 (25)	1 (5)	14 (70)	15 (75)	5 (25)	13 (65)	12 (60)	8 (40)	3 (15)	5 (25)	8 (40)

**Table 6 genes-14-01006-t006:** Genotyping pattern of *H. pylori* strains isolated from different types of raw poultry meat samples.

Type of Raw Milk Samples (Number of *H. pylori* Strains)	Genotyping Pattern (%)																		
	*s1am1a*	*s1am1b*	*s1am2*	*s1bm1a*	*s1bm1b*	*s1bm2*	*s1cm1a*	*s1cm1b*	*s1cm2*	*s2m1a*	*s2m1b*	*s2m2*	CagA+	CagA−	IceA1/IceA2	OipA+	OipA−	BabA2+	BabA2−
Chicken (9)	5 (55.55)	2 (22.22)	4 (44.44)	2 (22.22)	1 (11.11)	2 (22.22)	1 (11.11)	–	1 (11.11)	4 (44.44)	2 (22.22)	3 (33.33)	6 (66.66)	3 (33.33)	1 (11.11)	3 (33.33)	6 (66.66)	4 (44.44)	5 (55.55)
Turkey (7)	3 (42.85)	1 (14.28)	1 (14.28)	1 (14.28)	1 (14.28)	-	-	–	–	3 (42.85)	1 (14.28)	2 (28.57)	3 (42.85)	2 (28.57)	1 (14.28)	2 (28.571)	-	1 (14.28)	2 (28.57)
Ostrich (4)	1 (25)	–	-	-	–	–	-	–	–	2 (50)	–	1 (25)	1 (25)	-	–	1 (25)	-	-	1 (25)
Total (20)	9 (45)	3 (15)	5 (25)	3 (15)	2 (10)	2 (10)	1 (5)	-	1 (5)	9 (45)	3 (15)	6 (30)	10 (5)	5 (25)	2 (10)	6 (30)	6 (30)	5 (25)	8 (40)

## Data Availability

All data generated and/or analyzed during the current study are included in this published article. The datasets used and/or analyzed during this study are also available from the corresponding author on reasonable request.

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
