# Peer review of "Helicobacter pylori Strains Isolated from Raw Poultry Meat in the Shahrekord Region, Iran: Frequency and Molecular Characteristics"

_genes, 2023, doi:10.3390/genes14051006_

Round 1

Reviewer 1 Report

Dear Editor

Thank you for giving me the privilege to review this manuscript. The manuscript is about using virulent genes in Helicobacter pylori as genotypic indicators for H. pylori recovered from raw poultry. As laudable as the research look, because the impact of the information obtained from the study will be very useful for public health maintenance, it, however, suffers from a lot of grammatical, syntax, and organizational errors which need the authors revise to further improve their paper.

Author Response

Dear Editor,

On behalf of the authors, I would like to thank you and the reviewer for the careful and thorough reading of this manuscript and for the thoughtful comments and constructive suggestions, which absolutely help to improve the quality of our manuscript.

Our great appreciations for the Editor and Reviewers for their time and effort for giving these valuable remarks that have greatly improved the manuscript. After considering the comments point by point, we are looking forward to hearing positively about this manuscript.

Regarding

We have answered all the required questions raised by reviewers point by point as follows:

Review Report Form 1

Comments and Suggestions for Authors

Thank you for giving me the privilege to review this manuscript. The manuscript is about using virulent genes in Helicobacter pylori as genotypic indicators for H. pylori recovered from raw poultry. As laudable as the research look, because the impact of the information obtained from the study will be very useful for public health maintenance, it, however, suffers from a lot of grammatical, syntax, and organizational errors which need the authors revise to further improve their paper.

Reply: Thank you, proofreading of the manuscript was done, and it has been reworked.

Reviewer 2 Report

In the submitted manuscript, the authors tried to evaluate the distribution of H.pylori with key pathogenicity and AMR profiling in Iran raw poultry. The authors used the classical bacterial culture method and molecular biology tests to identify H.pylori. Then the presence of multi-virulence genes and antibiotic sensitivity was also characterized. In total, 20 H. pylori isolates confirmed by both PCR and biochemical testing were found in 320 poultry with the prevalence of multiple virulence factor genes, including the babA2, vacA, and cagA gene. Further AMR-sensitive testing showed high resistance to ampicillin, tetracycline, and amoxicillin. Overall, this manuscript is easy to follow. Moreover, ideas and key points are clear, and the data supports the conclusion. However, more than the current data and information may be needed for publication. Here, I list significant comments for authors' considerations to improve their manuscript.  

  1. Relationships of these 20 new H.pylori isolates. Based on the authors' data, these strains were isolated from chicken, turkey, and ostrich. Did the authors try to analyze the phylogenetic relationship between them and other historical isolates in Iran based on WGS and MLST analyses?  
  2. The authors only analyzed common drugs routinely used for H.pylori infection. However, this investigation should have included other AMR genes. Please use an AMR gene panel to detect and identify antibiotic resistance markers in these 20 isolates. Additionally, MIC should be determined. 
  3. Please provide references for primers used in this study.

Author Response

Dear Editor,

On behalf of the authors, I would like to thank you and the reviewer for the careful and thorough reading of this manuscript and for the thoughtful comments and constructive suggestions, which absolutely helped to improve the quality of our manuscript.

Our great appreciation to the Editor and Reviewers for their time and effort in giving these valuable remarks that have greatly improved the manuscript. After considering the comments point by point, we are looking forward to hearing positively about this manuscript.

Regarding

We have answered all the required questions raised by reviewers point by point as follows:

Review Report Form 2

Comments and Suggestions for Authors

In the submitted manuscript, the authors tried to evaluate the distribution of H.pylori with key pathogenicity and AMR profiling in Iran raw poultry. The authors used the classical bacterial culture method and molecular biology tests to identify H.pylori. Then the presence of multi-virulence genes and antibiotic sensitivity was also characterized. In total, 20 H. pylori isolates confirmed by both PCR and biochemical testing were found in 320 poultry with the prevalence of multiple virulence factor genes, including the babA2, vacA, and cagA gene. Further AMR-sensitive testing showed high resistance to ampicillin, tetracycline, and amoxicillin. Overall, this manuscript is easy to follow. Moreover, ideas and key points are clear, and the data supports the conclusion. However, more than the current data and information may be needed for publication. Here, I list significant comments for authors' considerations to improve their manuscript. 

Relationships of these 20 new H.pylori isolates. Based on the authors' data, these strains were isolated from chicken, turkey, and ostrich. Did the authors try to analyze the phylogenetic relationship between them and other historical isolates in Iran based on WGS and MLST analyses? 

Reply: Thanks. Phylogenetic analysis was not performed as the aim of this study was to investigate the frequency and molecular characteristics of Helicobacter pylori strains isolated from raw poultry meat in Iran and not to investigate relatedness between isolates.

The authors only analyzed common drugs routinely used for H.pylori infection. However, this investigation should have included other AMR genes. Please use an AMR gene panel to detect and identify antibiotic resistance markers in these 20 isolates. Additionally, MIC should be determined.

Reply: Thanks. The simple disk diffusion technique was used to study the antibiotic resistance pattern using the guidelines of the Clinical and Laboratory Standards Institute (CLSI). In addition, same antibiotic susceptibility method was used by Zohreh et al. (2022) to assess the antibiotic susceptibility pattern of H. pylori isolates. Antibiotic resistance genes were not detected because isolates will be further subjected to WGS. vacA, cagA, iceA, oipA and babA2 were detected in this study to describe the genotypic profile of circulating isolates.

  • Clinical and Laboratory Standard Institute In: Methods for Antimicrobial Dilution and Disk Susceptibility Testing of Infrequently Isolated or Fastidious Bacteria. Wayne, PA: National Committee for Clinical Laboratory Standards; 2015:M45.
  • Mashak, Z., Jafariaskari, S., Alavi, I., Sakhaei Shahreza, M., & Safarpoor Dehkordi, F. (2020). Phenotypic and genotypic assessment of antibiotic resistance and genotyping of vacA, cagA, iceA, oipA, cagE, and babA2 alleles of Helicobacter pylori bacteria isolated from raw meat. Infection and drug resistance, 257-272.

Please provide references for primers used in this study.

Reply: Thanks. References for primers were added in the manuscript (Table I)

Reviewer 3 Report

The manuscript entitled “Helicobacter pylori strains isolated from raw poultry meat: frequency and molecular characteristics” is straightforward and simple, well written and well organized by Authors.

Excellent and balanced introduction also supported by careful selection of supporting scientific literature. Well hooked on the problem from poultry meat to humans.

I would suggest that the Authors describe in a little more detail section 2.1 of the materials and methods specifying perhaps, not in the text of the manuscript, but in the rebuttal letter why some of the choices for identifying bacterial strains were made.

Correctly write the water formula with subscript numbers on line 126.

Good presentation of the results, but I would suggest editing Table 3 to make it more readable for readers.

The same consideration also applies to Table 5 and Table 6, which are unclear. It is probably just a layout error so that the categories are really not well interpreted.

Apart from these minor considerations, I would say that the manuscript is definitely well structured and worthy of publication after these minor changes.

Author Response

Dear Editor,

On behalf of the authors, I would like to thank you and the reviewer for the careful and thorough reading of this manuscript and for the thoughtful comments and constructive suggestions, which absolutely helped to improve the quality of our manuscript.

Our great appreciation to the Editor and Reviewers for their time and effort in giving these valuable remarks that have greatly improved the manuscript. After considering the comments point by point, we are looking forward to hearing positively about this manuscript.

Regarding

We have answered all the required questions raised by reviewers point by point as follows:

Review Report Form 3

Comments and Suggestions for Authors

The manuscript entitled “Helicobacter pylori strains isolated from raw poultry meat: frequency and molecular characteristics” is straightforward and simple, well written and well organized by Authors.

Excellent and balanced introduction also supported by careful selection of supporting scientific literature. Well hooked on the problem from poultry meat to humans.

I would suggest that the Authors describe in a little more detail section 2.1 of the materials and methods specifying perhaps, not in the text of the manuscript, but in the rebuttal letter why some of the choices for identifying bacterial strains were made.

Correctly write the water formula with subscript numbers on line 126.

Good presentation of the results, but I would suggest editing Table 3 to make it more readable for readers.

The same consideration also applies to Table 5 and Table 6, which are unclear. It is probably just a layout error so that the categories are really not well interpreted.

Apart from these minor considerations, I would say that the manuscript is definitely well structured and worthy of publication after these minor changes.

Reply: Thanks for the suggestion. We revised It.

Round 2

Reviewer 1 Report

The authors have improved the manuscript, but it still requires English editing. Most sentences are still unclear.  The authors should carry out more literature searches on Helicobacter because some ambiguous statements were made in the manuscript which must be clarified.

Author Response

Dear Editor, Many thanks to you and the reviewer for your time and effort in giving these valuable remarks that have improved the manuscript. After considering the comments point by point, we are excited to hear positively about this manuscript. Regarding   We have answered all the required questions raised by reviewers point by point as follows:   Review Report Form 1 Comments and Suggestions for Authors   The authors have improved the manuscript, but it still requires English editing. Most sentences are still unclear.  The authors should carry out more literature searches on Helicobacter because some ambiguous statements were made in the manuscript which must be clarified.   Reply: Thank you, A proofreading of the manuscript has been done.

Reviewer 2 Report

If the authors want to analyze the " frequency and molecular characteristics" of Helicobacter pylori. , the information about these isolates and their relationships with others should be included in this manuscript. 

Author Response

Dear Editor, Many thanks to you and the reviewer for your time and effort in giving these valuable remarks that have improved the manuscript. After considering the comments point by point, we look forward to hearing positively about this manuscript. Regarding   We have answered all the required questions raised by reviewers point by point:   Review Report Form 2 Comments and Suggestions for Authors If the authors want to analyze the " frequency and molecular characteristics" of Helicobacter pylori. , the information about these isolates and their relationships with others should be included in this manuscript.  Reply: Thank you, proofreading of the manuscript was done, and it has been reworked.
